# Efficacy of Carbamazepine and Its Derivatives in the Treatment of Bipolar Disorder

**DOI:** 10.3390/medicina57050433

**Published:** 2021-04-30

**Authors:** Anna Grunze, Benedikt L. Amann, Heinz Grunze

**Affiliations:** 1Psychiatrisches Zentrum Nordbaden, 69168 Wiesloch, Germany; anna.grunze@pzn-wiesloch.de; 2Centre Fòrum Research Unit, Institute of Neuropsychiatry and Addiction (INAD), Parc de Salut Mar, 08019 Barcelona, Spain; benedikt.amann@gmail.com; 3Department of Psychiatry Autonomous, University of Barcelona, CIBERSAM, 28029 Madrid, Spain; 4Psychiatrie Schwäbisch Hall, 74523 Schwäbisch Hall, Germany; 5Paracelsus Medical University, 90419 Nuremberg, Germany

**Keywords:** bipolar disorder, carbamazepine, depression, eslicarbazepine, maintenance, mania, oxcarbazepine, predominant polarity

## Abstract

*Background and Objectives:* This review is dedicated to the use of carbamazepine and its derivatives oxcarbazepine and eslicarbazepine in bipolar disorder and their relative strengths in treating and preventing new depressive or manic episodes. This paper will discuss the evidence of their efficacy relative to the polarity of relapse from controlled acute and maintenance/relapse prevention studies in bipolar patients. *Materials and Methods:* A Medline search was conducted for controlled acute and maintenance studies with carbamazepine, oxcarbazepine, and eslicarbazepine in bipolar disorder. In addition, abstracts reporting on controlled studies with these medications from key conferences were taken into consideration. *Results:* Information was extracted from 84 articles on the acute and prophylactic efficacy of the medications under consideration. They all appear to have stronger efficacy in treating acute mania than depression, which also translates to better protection against manic than depressive relapses for carbamazepine. Still, there is a paucity of controlled acute studies on bipolar depression for all and, with the exception of carbamazepine, a lack of long-term monotherapy maintenance data. For eslicarbazepine, the efficacy in bipolar disorder remains largely unknown. Especially with carbamazepine, tolerability issues and drug–drug interactions need to be kept in mind. *Conclusions:* Two of the medications discussed in this review, carbamazepine and oxcarbazepine, match Class A criteria according to the criteria proposed by Ketter and Calabrese, meaning acute antimanic efficacy, prevention of manic relapses, and not causing or worsening depression.

## 1. Introduction

The definition and use of the term mood stabilizers in the treatment of bipolar disorder (BD) remains a matter of discussion and disagreement. The classification suggested by Ketter and Calabrese [1] defines Class A mood stabilizers as stabilizing mood from *above* baseline and Class B mood stabilizers as stabilizing mood from *below* baseline, in which baseline means euthymia. It should be emphasized that the authors of this classification mainly focus on stabilizing mood, which includes the components of acute efficacy and maintaining this effect, but not primarily on prophylaxis, i.e., protection against recurrence of episodes in remitted patients.

Lithium (LI) might be a candidate for a mood stabilizer that comes close to fulfilling the criteria for both Classes A and B, although its relative strength in treating acute bipolar depression is not well evaluated [2,3] and results of double-blind, placebo (PLC)-controlled maintenance trials are inconsistent [4,5].

With the emergence of atypical antipsychotics as evidence-based treatment in BD, the role of anticonvulsants, with the exception of lamotrigine, has diminished, although in some countries, they are still popular and constitute at least a second line treatment for BD. Carbamazepine (CBZ), oxcarbazepine (OXC), and eslicarbazepine (ESL, available as eslicarbazepine acetate) belong to the dibenzazepine family of antiepileptic drugs and are all thought to primarily act as sodium channel and, to some degree, calcium channel blockers [6,7]. While CBZ blocks L-type calcium channels, OXC blocks N- and (or) P- and (or) R-type calcium channels. CBZ traditionally claims to be a mood stabilizer, at least in the sense of relapse prevention, despite the fact that it is not approved for this indication in most countries. Its neurochemical mechanisms of action support this assumption [8]. Theories on kindling mechanisms and behavioral sensitization have played a major role in translating antiepileptic efficacy into potential mood-stabilizing and prophylactic properties in bipolar patients [9,10,11]. Basic research also points toward some degree of similarities between LI and CBZ in the mechanisms of neuronal protection [12].

This narrative review now examines differential strengths and weaknesses of the three dibenzazepines CBZ, OXC, and ESL for treating and preventing the poles of BD, and whether these medications should be considered mood stabilizers according to the definition by Ketter and Calabrese [1].

## 2. Methods

We conducted a Medline search (1970 up to and including January 2021) for studies on mania, bipolar depression, maintenance, and relapse prevention with CBZ, OXC, and ESL to review their ability in treating and preventing affective episodes of one or both poles of the illness. We used the following MeSH terms (ALL FIELDS): carbamazepine OR oxcarbazepine OR eslicarbazepine AND (bipolar disorder OR mania OR manic depression). We retrieved 1443 results, with 193 reporting on clinical trials or meta-analysis. We also checked the literature lists of major textbooks on BD for additional references. In addition, data presented at key conferences or contributions to textbooks of which the authors were aware were also considered. In summary, 84 articles were identified that contributed useful information for the purpose of this article, of which 54 reported on specific clinical trials (including secondary analyses).

The Tolerability and Safety section in this paper is not based on a systematic literature search restricted to BD but is a narrative summary that also includes published data, especially from epilepsy.

### 2.1. Carbamazepine

Several older studies support antimanic properties of CBZ [13,14,15,16,17,18,19,20,21,22]. In these controlled investigations, CBZ was effective in an off-on-off design [23], superior to chlorpromazine [21], comparable to LI [15,16,17,22] and haloperidol [13], and statistically not worse than VPA [18]. Adding CBZ to haloperidol was superior to haloperidol plus PLC in a small number of patients with excited psychosis [14]. The latter finding appears questionable in so far that other investigators observed a worsening of psychosis when combining haloperidol with CBZ, most likely due to induction of the cytochrome P450 isoenzymes 2D6 and 3A4, resulting in a marked reduction in haloperidol plasma levels [24].

Three randomized double-blind PLC-controlled trials were identified using CBZ treatment in bipolar patients in different acute phases of BD. Two of these trials examined acutely manic and mixed patients (according to DSM-IV) and lasted for 3 weeks [25,26]. Singular analysis of the two identical studies as well as combined analysis [27] demonstrated superiority of CBZ for the primary (YMRS reduction from baseline to endpoint as well as several secondary outcomes). Of note, the first study also suggested improvement in depressive symptoms in mixed patients [25]. The third trial lasted for 12 weeks and compared CBZ alone and CBZ plus an herbal treatment (free and easy wanderer (FEW)) to PLC in 92, 96, and 47 Chinese patients, respectively, who were currently suffering from a depressive or manic phase [28]. In line with the two previous controlled studies, CBZ monotherapy produced significantly greater improvement on manic measures at week 2 through the endpoint compared to the placebo (PLC). CBZ monotherapy also yielded significantly higher clinical response rates than the placebo in mania (87.8% vs. 57.1%, *p* = 0.012).

Three controlled but methodologically limited and questionable studies pointed toward a possible antidepressant effect of CBZ in bipolar patients [29]. The first evidence stems from an off-on-off trial [23] reporting on 13 depressed patients, 4 of whom showed a good overall and 1 a moderate antidepressive response to CBZ, and a subgroup analysis of the controlled study by Zhang et al. [28] Both the CGI-S of depression and clinical response rates were significantly higher for CBZ than the placebo in bipolar depression (63.8% vs. 34.8%, *p* = 0.044) and mania (87.8% vs. 57.1%, *p* = 0.012). Maintenance of antimanic and antidepressive efficacy was observed in a 26-week extension of the study [30]. However, a recent network meta-analysis that included the study of Zhang et al. [28] as the only controlled study for CBZ labeled CBZ as ineffective regarding the primary efficacy outcomes (response and remission rate) in the treatment of bipolar depression. Comparing CBZ directly with PLC, the odds ratio for response was 2.01 (0.76–5.34 95%CI, *p* = 0.46) and for remission 0.90 (0.23–3.49 95%CI, *p* = 0.76) [31].

Several naturalistic and controlled trials have been conducted to evaluate CBZ’s efficacy in the long-term maintenance or prophylaxis of bipolar episodes [16,32,33,34,35,36,37,38,39,40,41,42,43]. Early studies were not conducted according to today’s methodological standards and were quite heterogenous in design; thus, meta-analytic comparison of CBZ vs. LI as prophylactic treatment yielded no conclusive results [44]. So, we need to look more closely at the individual trials to get an impression on the usefulness of CBZ in bipolar maintenance treatment.

Tracking unipolar and bipolar patients (*n* = 24) for up to three years, Ballenger and Post proposed first that CBZ might be useful for prophylactic treatment in BD [19]. Of the bipolar patients in their trial, two acutely depressed and two manic patients improved and maintained response. The same was true for one bipolar rapid-cycling patient and one patient with recurrent unipolar depression. Three more patients showed evidence of a partial prophylactic benefit.

In a 1-year, head-to-head PLC-controlled study with 22 patients, CBZ was found to be effective in 60% of the patients (compared to 22.2% with PLC) [33]. The prophylactic effect of CBZ against new manic episodes appeared evident, whereas CBZ’s efficacy in preventing depressive episodes could not be estimated due to the small sample size and a limited number of depressive episodes during the trial.

Watkins et al. [40] studied 37 patients with affective disorders, treated over an average period of 20 months with LI or 16 months with CBZ. Unfortunately, only 20 of the patients were bipolar. From today’s standpoint, the methodology was rather complex and questionable. The primary outcome was time in remission, composed retrospectively of time from the last episode to baseline and double-blind prospective treatment with CBZ or LI until emergence of a new episode. In this head-to-head investigation, LI was significantly better regarding time to relapse than CBZ. Unfortunately, the report supplies no detailed results about bipolar patients in this trial or about the polarity of affective relapses.

Forty responders of a double-blind, randomized 6-week acute study in which fifty-four manic patients received either CBZ or LI (27 each) were followed for 12 months of maintenance treatment [16]. Of the 20 patients on CBZ, 4 had a relapse into mania and 1 into depression, in comparison to 9 of 20 patients on LI, with 6 patients developing mania, 1 depression, and 2 a mixed state. The overall outcome was considered satisfactory for nine patients receiving CBZ and five taking LI. CBZ patients appeared slightly superior on every measure during the follow-up period and had a slightly lower number of depressive episodes.

Another 12-month, double-blind trial of CBZ vs. LI was carried out on 31 patients who were all previously stable on LI [35]. Fifteen patients were switched over to CBZ, and sixteen remained on LI. There was no statistically significant difference regarding the number of relapses into mania or depression during the observational period (eight on LI, six on CBZ). It is important to note that all but one of the patients who was switched from previous LI treatment to CBZ relapsed within the first 4 weeks. Thus, these relapses might mainly be due to the withdrawal of LI and have to be interpreted with care in respect of CBZ’s prophylactic efficacy. In addition, the severity of affective relapses was similar in both groups. Interestingly, only one patient in each group developed severe depressive symptoms, whereas six patients on LI and four patients on CBZ developed severe manic symptoms. This is somewhat contradictory to LI’s presumed tendency to be more protective against manic relapses and may be due to a selection bias with mainly mania-prone patients entering this study.

Simhandl et al. [41] published a randomized but open prophylactic study of CBZ with low or high serum levels in comparison to LI in both unipolar and bipolar patients. After 2 years, 58 of 84 patients had completed the study without any apparent differences in the overall efficacy of the prophylactic treatment. However, unipolar depressed patients showed no reduction in the duration of episodes with low CBZ levels (but with high levels of CBZ and LI). Of 58 patients who completed the trial, 7 patients with low CBZ levels, 9 with high CBZ levels, and 7 with LI had a relapse, and 8 (4 low CBZ, 2 high CBZ levels, and 2 LI) had to be hospitalized. Looking at the bipolar (*n* = 36) and unipolar subgroups (*n* = 22) separately and comparing the period 2 years before and 2 years on prophylactic treatment, there was a clear reduction in the number and duration of episodes (overall, manic, and depressed) and number of hospitalizations with CBZ in the bipolar group, with no apparent differences between LI and CBZ.

Denicoff et al. [36] compared the prophylactic efficacy of LI, CBZ, and the combination and tried to identify possible clinical markers of response. Fifty-two bipolar outpatients were randomly assigned in a double-blind design to an intended 1-year treatment with LI or CBZ, a crossover to the opposite drug in the second year, and then a third year on the combination. Of the evaluable patients, 13 (31.0%) of 42 failed to complete a full year of LI therapy due to lack of efficacy (4 due to a new manic episode, 9 due to depression), 13 (37.1%) of 35 withdrew from CBZ within the first year due to lack of efficacy (7 due to a new manic episode, 6 due to depression), and 7 (24.1%) of 29 withdrew from the combination therapy for lack of efficacy. The percentage of the evaluable patients who had marked or moderate improvement on the Clinical Global Impression (CGI) scale was 33.3% on LI, 31.4% on CBZ, and 55.2% on the combination treatment, which was a clear trend but not significantly different between groups. For a variety of measures, LI was more effective than CBZ in the prophylaxis of mania (number of dropouts due to lack of efficacy, percentage of time manic, number of days to the first manic episode), but all treatment modalities were more effective when compared retrospectively to the year before study initiation. The author also looked for correlates and predictors of response to the different treatments. Poor responsiveness to CBZ was associated with having more than 10 years elapsed between the onset of the first symptoms and study entry and with a history of rapid cycling.

A randomized, open multicenter study [37,45] with 144 bipolar and schizoaffective patients diagnosed according to ICD-9 compared the prophylactic efficacy of LI (*n* = 74) and CBZ (*n* = 70). LI was superior to CBZ in preventing recurrences and for the need of additional medication to treat relapses. In subsequent analyses, patients were re-classified according to DSM-IV criteria, causing a minor shift in categories and an increase in eligible patients. Dividing patients into classical bipolar I patients without mood-incongruent delusions and comorbidities (N = 67) and a nonclassical subgroup including all other patients (N = 104), LI was superior to CBZ in classical bipolar I disorder; however, a trend in favor of CBZ was also observed for the nonclassical group [46,47]. In addition, 51 (CBZ, *n* = 24; LI, *n* = 27) of the 171 patients who now fulfilled diagnostic criteria had to be re-hospitalized in this trial (personal communication, Greil and Kleindienst, 2004). The distribution of polarity of relapses was similar between patients on LI and CBZ, with slightly more patients experiencing a depressive than manic or mixed relapse.

Another analysis [38] focused on the whole bipolar I subgroup (N = 114 of 171). A statistically significant higher dropout rate was found in the CBZ compared to the LI group. However, again there was no difference between groups (LI, *n* = 58; CBZ, *n* = 56) with respect to the affective symptomatology at first re-hospitalization. In the LI group, 38% developed depressive, 31% manic, and 31% other (such as schizoaffective) symptoms. Of the patients on CBZ, 37% showed depressive, 21% manic, and 42% other symptoms. Similarly, in a sub-analysis of bipolar II patients (*n* = 57; LI, *n* = 28; CBZ, *n* = 29) no significant differences in the number of hospitalizations or recurrences were found between groups [48]. In the bipolar II group, seven LI patients had to be re-hospitalized, one with a new hypomanic episode (14%), two with depressive episodes (29%), and four with other symptoms (57%). In the CBZ bipolar II group, three patients had to be re-hospitalized, two with depression (67%) and one with other symptoms (33%). Clearly, the small number of patients (*n* = 3) has to be taken into account and any conclusions have to be drawn with care (personal communication, Kleindienst and Greil, 2004).

Hartong et al. [39] studied 94 patients with at least two episodes of BD during the previous 3 years. Previous treatment with LI or CBZ had not exceeded a total of 6 months during their lifetime. Remitted patients were randomly assigned to CBZ or LI monotherapy at entry into this 2-year double-blind study. Alternatively, patients were also eligible for randomization during an acute index episode prior to entry into the study. The latter enrichment of the study with acute responders was a result of a protocol change in order to enhance recruitment. Only part of the patients in this study were therefore not preselected for prophylactic responsiveness to either treatment. Still, bearing in mind the relatively long duration of this study (2 years), it comes close to being a true prophylaxis trial. On LI treatment, 12 of 44 patients developed an affective episode, compared with 21 of 50 on CBZ treatment. Episodes on LI treatment occurred almost exclusively during the first 3 months of the trial. CBZ, however, carried a constant risk of an episode of about 40% per study year. The efficacy of LI was superior to that of CBZ in patients with a (hypo)manic index episode that had not been treated with a study drug during the index episode (*p* < 0.01) and also in patients with prior hypomanic but no manic episodes (*p* < 0.05). The proportion of patients who dropped out was slightly higher among those taking LI (16/44) compared with those taking CBZ (13/50). In addition, 16 of 44 patients (36%) on LI treatment completed the 2 years with no episode, compared with 16 of 50 (32%) on CBZ treatment. In the LI group, 4 patients dropped out due to a new (hypo)manic episode and 12 due to a new depressed episode, whereas in the CBZ group, 10 patients experienced a new (hypo)manic and 11 a new depressive episode. The authors did not test for significance, but it appears that the mania-prophylactic properties of LI are stronger, whereas no difference between manic and depressed relapses became apparent for CBZ.

Several other points of this study are remarkable. First, the survival plot suggests different mechanisms of action, with LI being truly protective in those surviving the first months, whereas CBZ seems to have a more short- to medium-term prophylactic effect. Second, LI appears to be effective also in patients without the pre-requisite of acute responsiveness in an acute manic episode, which appears different for CBZ.

Table 1 supplies an overview of the relevant controlled prophylaxis studies with CBZ.

Superiority of LI over CBZ (and VPA) was also demonstrated in a large naturalistic head-to head study of these three mood stabilizers. Patients were followed up to 124 months until recurrence of an episode or study termination. The median unadjusted survival time was 36 months for patients taking VPA, 42 months for patients taking CBZ, and 81 months for patients taking LI [42]. Finally, a retrospective analysis of charts of 129 patients with a mean duration of CBZ use of 10.4 ± 5.2 years was also in favor of the prophylactic efficacy of CBZ. Mirror-image comparison of the frequencies of admission per year before and after CBZ treatment initiation showed an obvious reduction, from 0.33 ± 0.46 to 0.14 ± 0.30, and 63 (48.8%) patients had no further episodes during the 10-year follow-up period [43].

Accepting that LI should be the first-choice prophylactic treatment for BD (in the absence of contraindications), the question of which medication to add in the case of an insufficient response arises. Among antiepileptic mood stabilizers, CBZ and VPA are potential candidates, especially in patients with a manic polarity of episodes. Missio and colleagues conducted a randomized open trial comparing LI plus VPA versus LI plus CBZ in young BD-I patients (LICAVAL study) [55]. They found no differences between treatment arms in the primary outcome (Clinical Global Impression scale modified for use in BD (CGI-BP)) or in any secondary outcome. However, there were more tolerability issues in the LI + VPA group. In addition, the LI + VPA group gained weight (+2.1 kg), whereas the LI + CBZ group had a slight weight loss (−0.2 kg). Keeping in mind that weight gain is one of the most frequent reasons for non-adherence to medication [56], addition of CBZ might be a better option than VPA in patients not fully responding to LI.

In conclusion, CBZ demonstrates acute antimanic properties [57], whereas there is only a weak hint of an acute antidepressive effect. The latter is mainly due to a lack of sufficient data from studies employing sufficient numbers of patients. Methodological shortcomings also limit conclusions about the prophylactic efficacy of CBZ. Considering a differential efficacy to prevent depressive and manic episodes, CBZ seems comparable to LI. Similar to LI, CBZ seems to be more effective in preventing manic relapses (apart from the study by Lusznat [16]; see discussion below), but in direct comparison to LI, CBZ’s mania-prophylactic effect appears less pronounced [36]. However, the majority of prophylactic trials with CBZ included only small numbers of patients or had an insufficient observation period of 1 year. In addition, its long-term protective properties remain to some degree uncertain due to the lack of a sufficient number of PLC-controlled trials.

The study and sub-analyses by Greil and colleagues comparing randomized treatment with CBZ and LI over 2.5 years supplied valuable information in these respects. CBZ clearly prolongs the time to an immediate relapse in patients who have been recently ill. However, considering the outcome of the study by Hartong [39] in treatment-naive patients, where CBZ was superior to LI for the first 8 months but performed worse after 2 years, carrying a constant recurrence risk of 40% per study year, and the better outcome with CBZ in patients pre-selected for acute responsiveness [16], it is probably incorrect to assign CBZ prophylactic efficacy in its truest sense. Taking all trials of CBZ into account, it appears that it is efficacious in stabilizing bipolar patients from above baseline, i.e., as a Class A agent, but evidence is lacking that it is also effective in stabilizing from below baseline as a Class B medication, which means exhibiting antidepressive efficacy in acute bipolar depression.

### 2.2. Oxcarbazepine

Although frequently used in patients not tolerating CBZ, the scientific evidence for efficacy in BD is considerably less for OXC than for CBZ.

Most of the work on OXC in BD, especially in acute mania, was performed by Emrich et al. in the early 1980s [58,59]. The antimanic effect of OXC appeared comparable to the one of VPA in these early studies. In a double-blind comparative trial with haloperidol by Müller and Stoll [60], no difference in efficacy was observed between OXC and haloperidol. This impression was also supported by another study by Emrich [61]. With similar efficacy, the tolerability of OXC was significantly better, with the incidence of side effects 3.5 times higher in the haloperidol group compared to the OXC group [61]. These studies were also comprehensively reviewed by Dietrich et al. [62]. Afterward, a study by Hummel et al. [63] supplied further supporting evidence for the antimanic efficacy of OXC, using an on-off-on design. In this study, OXC monotherapy was effective in patients with mild-to-moderate mania (Young Mania Rating Score (YMRS) < 25) only but not in severely manic patients.

So far, no controlled studies have been reported for OXC in bipolar depression [64]. A retrospective chart review by Ghaemi et al. [65] suggests that OXC may possess mild-to-moderate efficacy in bipolar depressed patients. A short-term study reported that OXC add-on therapy is not only effective in treating mania but also effective in treating acute bipolar depression [66]. Benedetti and colleagues conducted a longer-term (8 weeks of acute treatment with a subsequent 4- to 12-month follow-up), open, add-on study in four patients with mania, eight patients with bipolar depression, and eight patients with a mixed index episode: 61% met the criteria for response according to their CGI scores at week 8, and a substantial proportion (66.3%) of the 8-week trial responders maintained a satisfactory mood stabilization during the follow-up [67]. However, the impression of an antidepressive effect of OXC has not been verified by controlled studies so far.

For maintenance in BD, a meta-analysis in 2008 identified two studies that, however, did not qualify as sufficient and methodologically rigorous evidence [68]. Cabrera and colleagues [69] conducted a comparator study against LI for up to 16 months, however, in a small number of patients (*n* = 19). Wildgrube [70] conducted a similar 31-month study of maintenance treatment in BD and schizoaffective disorder. The results of both studies pointed toward a decrease in both manic and depressive relapses with OXC. Additional evidence for efficacy in maintenance/prophylactic treatment stems from case series and open studies. Munoz [71] investigated the mood-stabilizing effect of OXC 300–2400 mg/d as adjunctive treatment over 12 weeks. Improvement according to pre-defined criteria was observed after 3 weeks in 71% of patients entering this study manic and all patients entering this study depressed, with 60% of the responders of the manic group remaining euthymic through the remaining 9 weeks of the study and all of the responders in the depressed group remaining stable for the remainder of this study. Nasr and Casper (2002) [66] conducted a further open but prospective review in 87 patients with mood disorders, including 28 patients with BD. With the addition of OXC, a significant improvement in the CGI severity-of-illness score was observed in all patients. The greatest improvement was seen in bipolar-II patients with predominant hostility. In the study of Reinstein [72], 57 patients with bipolar or schizoaffective disorder were stabilized with VPA. Half of them were switched to OXC after 10 weeks of treatment and followed up for another 10 weeks. Compared to patients remaining on VPA, mood stability was maintained to a similar degree. As a secondary outcome, it was noted that 47% of patients on VPA gained weight compared to only 26% of patients on OXC and a substantial proportion (70%) of patients lost weight after being switched from VPA to OXC.

The promising results of these early, mostly open studies encouraged Vieta and colleagues [73] to conduct a randomized, PLC-controlled prophylaxis study of oxcarbazepine add-on to LI. The primary efficacy variable was the length of the remission period assessed by means of the YMRS and Montgomery–Asberg Depression Rating Scale (MADRS). The outcome of the patients on oxcarbazepine was not significantly different (*p* = 0.315) from that of patients on PLC during the 1-year follow-up. However, the authors described a trend toward prophylactic efficacy against depressive recurrences and a significant positive effect on the prevention of impulsivity. Recent meta-analytic reanalysis of this study, however, demonstrated a prophylactic effect of LI plus OXC, with a strong trend not only toward prevention of new depressive episode (RR 0.294, 95% CI 0.086–1.002) but also against new manic, hypomanic, or mixed episodes (RR 0.301, 95% CI 0.101–0.899) [74].

In conclusion, there is some evidence of acute antimanic effects of OXC, at least in mild-to-moderate manic patients but probably not in severely ill patients. Data on the treatment of bipolar depression are virtually absent and for prophylactic efficacy scarce but might bear some promise. According to the criteria by Ketter and Calabrese [1], OXC would, similar to CBZ, qualify rather as a Class A agent, albeit the evidence base is weaker than with CBZ. However, OXC may be a potential candidate drug for patients who previously responded well to CBZ but had problems with tolerability or drug interactions in combination treatment.

### 2.3. Eslicarbazepine

A first case report suggestive the antimanic efficacy of ESL was published in 2012 [75]. At this time, ESL had already been studied in two 3-week double-blind, randomized, PLC-controlled trials in acute mania, one study using a flexible dose (up to 2400 mg/d) [76] and the other a fixed dose design (up to 1800 mg/d), both followed by a recurrence prevention study for a minimum of 6 months. While the fixed dose study was discontinued prematurely due to slow recruitment, in the flexible dose study, ESL was numerically but not significantly better than PLC in the primary outcome (change in the total YMRS score from baseline to endpoint). However, in secondary outcomes, significantly more patients achieved full remission with ESL than with PLC. Thus, a mild-to-moderate antimanic effect of ESL appears likely, although it is not proven. In the 6-month recurrence prevention study with three different doses of ESL (300, 900, 1800 mg/d), less than half of the patients experienced worsening (increase in YMRS ratings) [77].

So far, no controlled data have been published for the efficacy of ESL in bipolar depression. Given the limited evidence, the positioning of ESL in the treatment of BD remains unclear.

### 2.4. Use of Carbamazepine, Oxcarbazepine, and Eslicarbazepine in Special Populations with BD

Whereas the use of CBR, OXC, and, to a lesser degree, ESL is well established across the life span in epilepsies, data on their use in children, adolescents, and the elderly with BD are sparse. For children and adolescents, they are not indicated for BD in most countries under the age of 18 [78]. In the absence of confirmative studies, evidence rests with case reports and open trials. An 8-week, open-label, prospective trial of CBZ extended-release monotherapy trial in 27 children with bipolar spectrum disorder found a statistically significant, but only modest, level of improvement in the mean YMRS score [79]. The best evidence for efficacy of CBZ in juvenile BD stems from a 26-week, open-label study in 60 children (ages 10–12) and 97 adolescents (ages 13–17) with an acute manic or mixed episode. At study end, the most prevalent dose of CBZ extended-release capsules was 1200 mg. The YMRS score decreased from a mean of 28.6 ± 6.2 at baseline to a mean of 13.8 ± 9.4 (*p* < 0.0001) at the endpoint. However, in the absence of an established comparator or placebo, the result is difficult to interpret, especially given the long trial duration where a high rate of natural remission can be expected. The safety and tolerability profile were not different from the one in adults with BD.

Whereas there are no reports on ESL in juvenile BD, the quality of evidence for OXC appears better than for CBZ; however, the evidence is negative. One hundred sixteen outpatients aged 7 to 18 years with BD-I, manic or mixed, were randomly assigned to receive 7 weeks of double-blind, flexible dose treatment with OXC (900–2400 mg/day) or a placebo. As a result, OXC was not significantly superior to the placebo in this study [80].

Among other mood stabilizers, CBZ is quite frequently prescribed in old-age BD [81]. As far as evidence is concerned, no controlled study has been conducted with CBZ, OXC, or ESL exclusively in this population. Naturalistic data suggest comparable efficacy of CBZ in old- age BD. Sanderson and colleagues [82] compared retrospectively the length of stay and symptom improvement in elderly BD inpatients receiving monotherapy with lithium, valproate, or CBZ and found no significant differences across the groups. The mean ± SD change in Global Assessment of Functioning (GAF) scores during hospitalization was 28.8 ± 11.8 for patients who received carbamazepine, 29.9 ± 15.8 for patients who received lithium, and 35.2 ± 10.9 for patients who received valproate. The paper did not specify how many patients were admitted for mania, mixed states, or depression.

### 2.5. Tolerability and Safety of CBZ, OXC, and ESL

The use of CBZ both in epilepsies and in BD, however, is often associated with tolerability issues [83]. With higher dosages, these typically include blurred vision or diplopic images, nystagmus, confusion, hyponatremia, and urinary retention. Hyponatremia is a common adverse effect not only of CBZ but also of OXC and ESL. Clinical signs of hyponatremia include nausea with vomiting, fatigue, lethargy, headache or confusion, muscle cramps or spasms, irritability, restlessness, or weakness. With a further, especially rapid, drop in sodium levels, seizures, loss of consciousness, and finally coma may occur. Fortunately, ECG abnormalities with hyponatremia are mostly unspecific and rarely of clinical significance [84]. As hyponatremia due to medication usually develops slowly and is rather chronic but acute, a change in medication, oral sodium substitution, and/or restriction of fluids are usually sufficient treatments. If with low sodium levels (<125 mmol/L), salt substitution is needed, it should be done gradually to avoid osmotic demyelination syndrome (ODS) [85]. The usually recommended rate is not more than 10 mmol/L/day for both acute and chronic hyponatremia [86].

Within the therapeutic serum range, severe side effects such as agranulocytosis, aplastic anemia, and allergies (rarely Steven–Johnson syndrome (SJS) and toxic epidermal necrolysis (TEN)) may occur more frequently than with other mood stabilizers [83]. Carriers of the human leukocyte antigen (HLA) B*15:02 (common in people of Asian origin) and HLA-A*31:01 bear the highest risk of SJS and TEN induced by CBZ or OXC [87,88]. Neurotoxic side effects are likely related to CBZ 10,11-epoxide, the first metabolite of CBZ. When combining CBZ with phenytoin, phenobarbital, primidone, and valproic acid, the ratio of CBZ 10,11-epoxide to CBZ increases and toxicity may easily occur with CBZ serum levels within the therapeutic range [89]. As CBZ is metabolized by the liver, transient hepatic enzyme elevations occur at a frequency of 5–15%, whereas fulminant hepatitis is exceedingly rare. In addition, transient leukopenia may occur in up to 10% of patents during the first weeks of treatment, which may persist in 2% of cases [90]. Agranulocytosis or aplastic anemia induced by CBZ is rare (approx. 1 in 100,000 patients); however, special caution should be applied when combining CBZ with clozapine or mirtazapine, because of possible synergistic effects on bone marrow suppression [29]. Treatment with CBZ also has limitations because of induction of cytochromes (CYP P450 2D6, 3A4) and glucuronizing enzymes (as well as P-glycoprotein) and subsequent interaction with the metabolism or bioavailability of other drugs. CBZ itself is mainly metabolized by CYP 3A4; however, of special clinical relevance is the induction of CYP2D6 that metabolizes about 25% of all medications, including several antipsychotics and antidepressants. Table 2 lists medication frequently used in psychiatry and frequent comorbid somatic disorders that are affected by CBZ; a more comprehensive list also including more drugs for non-psychiatric indications can be found at https://www.mayoclinic.org/drugs-supplements/carbamazepine-oral-route/precautions/drg-20062739?p=1 (accessed on 16 April 2021). In contrast, some medications may increase CBZ plasma levels, causing toxicity, such as fluoxetine, fluvoxamine, nefazodone, and VPA. In addition, CBZ induces its own metabolism, demanding adaptions of dosages.

Different from CBZ, its analogue OXC follows a different metabolic pathway. OXC is only a mild inducer of CYP3A4 and UDP-glucuronosyltransferases (UGTS) and a moderate inhibitor of CYP2C19. OXC is mainly (about 55%) reduced by a cytosolic arylketone reductase to its 10-mono-hydroxy derivatives ESL and, to a much lesser extent, R-licarbazepine. Unlike CYP 3A4, arylketone reductase is not an inducible enzyme. Thus, whereas CBZ induces its own metabolism, this does not occur with OXC [91]. ESL appears to be the pharmacological active agent in epilepsies. It is a mild inducer of CYP3A4 and UGTS and a moderate inhibitor of CYP2C9 and CYP2C19. Different from CBZ, both OXC and ESL do not induce their own metabolism. ESL is metabolized via glucuronide conjugation (UGT1A4, UGT1A9, UGT2B4, UGT2B7, UGT2B17) and excreted by the kidneys. Thus, the chance of possible interactions with other psychotropic drugs is much lower for OXC and ESL than with CBZ [92]. This makes both OXC and ESL potential candidate drugs for use in BD, with a better tolerability profile than CBZ. Studies both in epilepsies and in BD with ESL reported only mild-to-moderate intensity of side effects [77,93,94,95]; however, meta-analysis of all PLC-controlled studies of OXC and ESL showed a significantly higher incidence of serious adverse events (AEs) compared to PLC. The majority of AEs of ESL were related to the vestibulo-cerebellar system, but hyponatremia and rash also occurred with higher dosages (1200 mg/d) [96]. Thus, ESL and OXC are not without side effects, and especially hyponatremia may occur more often with OXC and ESL than with CBZ [97]. However, an advantage of all three dibenzazepines is a lower propensity of weight gain compared especially to LI, VPA, and several atypical antipsychotics [98]. For detailed information about the metabolism, pharmacokinetic effects, and pharmacodynamic effects of OXC and ESL, we refer the reader to the comprehensive literature (e.g., [99]).

Teratogenicity is an issue that clearly limits the use of CBZ in women of child-bearing potential (WOCBP). Therapeutic dosages of CBZ decrease the levels of contraceptive steroids and may also increase breakthrough bleeding and permit ovulation when using low-dose oral contraceptives. Similar findings have been reported for OXC [100] and ESL [101]. Concerning teratogenicity, the risk for CBZ is well known [102] and estimated to be 2.2% for major congenital malformations [103]. Data for OXC appear to be not different from the rate that would be expected in the general population [104], and for ESL, the database is still too small for any conclusions about safety in pregnancy [105]. Breastfeeding after post-partum re-introduction of CBZ is possible, as it does not appear to adversely affect infant growth or development, and breastfed infants of mothers on CBZ monotherapy had even higher IQs and enhanced verbal abilities than nonbreastfed infants at 6 years of age in one study [106].

Another issue that needs to be addressed in the long-term treatment is decreasing bone mineral density causing osteoporosis that can occur with several CYP P450-inducing antiepileptics, most frequently, however, with CBZ [107] inducing the metabolism of vitamin D [107,108]. Therefore, guidelines recommend prophylactic prescription of vitamin D [109] and regular checks of vitamin D levels as well as regular bone densitometry. In the case of a vitamin D deficiency, substitution alone or in combination with calcium is strongly recommended, as both CBZ and OXC can also induce secondary hyperparathyroidism.

The relative low risk of weight gain and other metabolic issues with CBZ, OXC, and ESL, however, are advantageous for long-term treatment, especially in patients who had these problems when taking lithium, VPA, or second-generation antipsychotics [29].

There are no published data for the cost-effectiveness of OXC and ESL in BD. In general, cost-effectiveness analyses for classical mood stabilizers are scarce in comparison to second-generation antipsychotics [110]. In epilepsies, CBZ is considered a cost-effective treatment choice [111]. However, a systematic review of randomized or quasi-randomized controlled trials of BD maintenance therapies followed by a mixed-treatment comparison found that CBZ is not an effective maintenance treatment for bipolar-I disorder [112]. With limited financial resources of health systems and generic availability, however, CBZ is still an attractive choice also in BD, with daily treatment costs (600 mg/d) of 1.75 US$. For OXC and ESL, which are only available as brands, daily treatment costs of 1200 mg/d are 37 and 101 US$, respectively.

## 3. Discussion

Especially for long-term treatment, tolerability and safety become almost as important as efficacy. Whereas some adverse drug reaction, e.g., allergic rash, manifest quite early in treatment, others, e.g., hyponatremia, osteoporosis, or hepatotoxicity, may develop over time. Thus, it is important to monitor bipolar patients taking CBZ with a similar rigor as patients on other antiepileptic mood stabilizers or second-generation antipsychotics. The German S3 guideline for bipolar disorder recommends to have a thorough physical examination at treatment initiation and every year and to monitor at the beginning and every six months the patient’s weight and the following lab parameters: hepatic and renal function, full blood count, and CBZ serum level [113]. Similar recommendations have been made by the International Society for Bipolar Disorder (ISBD) [114]. Clearly, in the first weeks after start of treatment, more frequent lab controls are indicated. Although not part of official bipolar guidelines, a similar monitoring scheme should apply to OXC and ESL.

The choice of alternative acute treatment has widened considerably over the past decade, especially with the emergence of second-generation antipsychotics; however, for maintenance and prophylactic treatment, CBZ still constitutes an option worth to consider. Especially, patients with atypical features and forms of BD do not respond as well to lithium, long-term data for VPA are less convincing than for CBZ, lamotrigine might be insufficient to protect against new manic episodes, and the long-term use of several second-generation antipsychotics is associated with weight gain and metabolic issues. In addition, except olanzapine, no second-generation antipsychotic has, so far, proven efficacy in a prophylaxis study, but only in continuation/maintenance trials in samples enriched for acute response and mostly lasting no longer than six months [115,116,117]. Finally, some groups, e.g., the elderly, may have medical safety issues prohibiting lithium therapy (renal impairment) or use of antipsychotics (cerebrovascular disorder, emerging dementia). Thus, the authors feel that there is still a place for CBZ, and possibly also OXC, in the treatment of BD.

## 4. Conclusions 

A categorization of mood stabilizers, as suggested by Ketter and Calabrese, may be superficial, considering the complexity of illness; however, it may be of clinical utility. Two medications discussed in this review, CBZ and OXC, match Class A criteria, meaning acute antimanic efficacy, prevention of manic relapses, and not causing or worsening depression. There is no convincing data for either CBZ or OXC in the acute treatment of depression or the prevention of depressive relapses, and the duration of most controlled studies is insufficient to allow any conclusions about prophylactic efficacy over years. Clearly, more long-term observational data, similar to those for LI, are needed for these medications. For ESL, evidence is too scarce to allow for any statement about its effectiveness in BD. 

## Figures and Tables

**Table 1 medicina-57-00433-t001:** Prophylaxis studies with CBZ in BD.

Study	Comparator	Randomized	Blinded	N (CBZ)	N (Comparator)	Duration (Months)	Response Rate CBZ (%)	Response Rate Comparator (%)
Okuma et al. [33]	Placebo	Yes	Yes	12	10	12	60	22
Post et al. [32]	Placebo	M	Yes	7	7	19	86	-
Kishimoto and Okuma [49]	LI	C	-	18	18	≥24	CBZ > LI	
Placidi et al. [50]	LI	Yes	Yes	20	16	≤36	67	67
Watkins et al. [40]	LI	Yes	Yes	19	18	18	84	83
Lusznat et al. [16]	LI	Yes	Yes	16	15	≤12	56	29
Elphick et al. [51]	LI	C	Yes	8	11	9	38	73
Bellaire et al. [52]	LI	Yes	-	46	52	12	CBZ = LI	
Mosolov [53]	LI	Yes	-	30	30	≥12	73	70
Di Costanzo [54]	LI	Yes	-	8	8	≤60	LI + CBZ > LI	
Coxhead et al. [35]	LI	Yes	Yes	13	15	12	54	47
Greil et al. [37]	LI	Yes	-	70	74	30	45	65
Denicoff et al. [36]	LI	C	Yes	46	50	12	33	55
Simhandl et al. [41]	LI	Yes	-	58	26	24	CBZ = LI	
Hartong et al. [39]	LI	Yes	-	50	44	24	58	73
**Total (weighted means)**				421	394		54	64 (LI) 22 (Placebo)

CBZ: carbamazepine; LI: Lithium; M: mirror image; C: crossover. For four studies, detailed response rates have not been reported, but only superiority/equality of drugs tested for various outcomes. Table adapted and updated from [29].

**Table 2 medicina-57-00433-t002:** Medications where CBZ pharmacokinetically decreases the serum levels.

Class of Medication	Drug
**Antidepressants**	Bupropion
	Citalopram
	Mirtazapine
	Sertraline
	Tricyclic antidepressants
	MAO inhibitors (used in depression or Parkinson’s disease: iproniazid, moclobemide, phenelzine, rasagiline, selegiline, tranylcypromine)
**Antipsychotics**	Aripiprazole
	Clozapine
	Haloperidol
	Lurasidone
	Olanzapine
	Risperidone
	Ziprasidone
**Anxiolytics/Sedatives**	Alprazolam
	Clonazepam
**Anticonvulsants**	CBZ (autoinduction)
	Lamotrigine
	Oxcarbazepine
	Phenytoin
	Primidone
	Tiagabine
	Topiramate
	VPA
	Zonisamide
**Stimulants**	Methylphenidate
	Modafinil
**Analgesics**	Fentanyl
	Methadone
	Buprenorphine
	Tramadol
**Anticoagulants**	Warfarin
**Muscle relaxants**	Pancuronium
	Rocuronium
	Vecuronium
**Steroids/Hormones**	Dexamethasone
	Mifepristone
	Oral contraceptives
	Prednisone
	Thyroid hormones
**Immunosuppressants**	Cyclosporin
	Sirolimus
	Tacrolimus
**Antimycotics/Antibiotics**	Caspofungin
	Doxycycline
	Voriconazole
**Antivirals**	Delavirdine
	Protease inhibitors
**Others**	Nimodipine
	Quinidine
	Repaglinide
	Theophylline

Bold illustrating that these are headings (groups) whereas the right column are individual drugs.

## Data Availability

All data given in this review have been previously published and are in the public domain.

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
