# Peer review of "Efficacy of Carbamazepine and Its Derivatives in the Treatment of Bipolar Disorder"

_medicina, 2021, doi:10.3390/medicina57050433_

Round 1
Reviewer 1 Report
Interesting article on the use of carbamazepine in bipolar disorder.
Carbamazepine is a drug that has been used for many years, so analyses of its use are important.
However, the paper reads tediously. There is a lack of tabular summaries. Above all, there is a lack of conclusions, and the discussion should be expanded. In addition, there should be a paragraph (summary) of carbamazepine interactions.
Author Response
We thank the reviewer for his balanced and informative review. Changes according to his/her remarks are marked in the revised manuscript in blue. As there is some overlap with suggestions of reviewer 2, e.g., content of discussion/summary, please see also changes marked in green.
As suggested, we extended paragraphs dealing with CBZ interaction, the Discussion and Summary section. We now moved the whole issue of tolerability and safety away from the introduction assigning it a chapter on its own. Following suggestions of the second reviewer, we also extended sections on adverse effects and special patient groups. We agree with the reviewer that tabular overviews contribute much to better readability and make it less tediously, so we added a tabular overview of prophylaxis trials with CBZ and the most important interactions of CBZ in psychiatry. We hope that this new version is now satisfactory.

Reviewer 2 Report
The work consists of a narrative review on the efficacy of carbamazepine and its derivatives in the treatment of bipolar disorder. It is of clinical interest, since carbamazepine is used less and less, at the expense of a massive use of atypical antipsychotics. Therefore, it is appropriate to synthesize the evidence of its efficacy and other issues (tolerability, security). The review is complete, well written, and meets scientific quality standards. It may be of interest to clinicians and researchers in the field of bipolar disorder.
I would propose the following changes:
- The title “Spectrum of efficacy of carbamazepine and its derivatives in bipolar disorder” is a bit confusing, due to the well-known concept of “bipolar spectrum”. The reader could interpretate that it´s a review on the efficacy of carbamazepine in bipolar spectrum. I´d suggest: “Efficacy of carbamazepine and its derivatives in the treatment of bipolar disorder”
- I would include aspects of tolerability and interactions of carbamazepine in the Abstract. They are very relevant aspects for its clinical management.
- In the Introduction and Results, I would address the important issue of the teratogenicity of the reviewed drugs.
- In the Results section, there are often thoughtful comments that should be included in Discussion. This contrasts with a very short Discussion section. In brief, I would recommend cutting Results and lengthening Discussion.
- In the Discussion, I would address the role that the main international Clinical Practice Guidelines give to the drugs reviewed. I would discuss the current trend towards the massive use of antipsychotics in bipolar disorder, with controversial evidence. Reference: Escudero et al. Second Generation Antipsychotics Monotherapy as Maintenance Treatment for Bipolar Disorder: a Systematic Review of Long-Term Studies. Psychiatr Q. 2020 Dec;91(4):1047-1060. doi: 10.1007/s11126-020-09753-2. PMID: 32651765.
- It would be interesting to add data on the evolution of the worldwide use of carbamazepine in the treatment of bipolar disorder in the last 40 years.
- I would add some comment on the cost-effectiveness of the reviewed drugs.
- I would add a specific section on the use of these drugs in special populations (children and adolescents, the elderly).
- In the Discussion, I would conclude with a summary of the strengths and weaknesses of carbamazepine (and its derivates), with a comparison with lithium, valproate, lamotrigine, and antipsychotics.
- Hyponatremia associated with the use of carbamazepine is a relevant clinical problem. I would go into this topic in greater depth. In general, I believe that the article can be very valuable if it is specially formatted for the clinician, offering an aid in making therapeutic decisions in bipolar disorder.
Author Response
Thank you for your balanced and informative review. Changes in the manuscript following your suggestions have been marked in green.
Title: We changed it according to your suggestion
Abstract. We added a short notice about interaction and tolerbility. Due to the regulations (300 words max) we could not extend on it more in detail.
We now moved the whole issue of toleraability and safety away from the introduction assigning it a chapter on its own. We gave more detailled information on teratogenicity and interaction with oral contraceptives to put it into a clinical perspective. However, as the focus of this article is on the efficacy of CBZ, OXC and ESL, we decided not to extend on it in the results as we did not systematically search for safety and tolerability issues.
Unfortunately, we have no access to data on the usage of CBZ in the last 40 years. However, these data might be inconclusive for Bipolar Disorder as the fast majority of usage is in epilepsy, and additionally confounded by the bulk of new medication introduced in epilepsy over the past decades, leading to a decline of use of CBZ also in this major indication.
Unfortunately, there are no published cost-effectiveness data for OXC and ESL in BD, only for CBZ which we included in the paper.
We also gave more detailled information on symptoms and treatment of hyponatremia, other long term adverse effects and CBZ treatment in special populations. We also extended on the Discussion and Summary, detailing pro and cons of CBZ in comparison to other mood stabilizers.
We hope that this changes/addition are satisfactory to the reviewer, they clearly contributed to improving the educational value of the paper.

Round 2
Reviewer 1 Report
The reviewer's comments have been taken into account. The paper in its current form meets the requirements of the journal and may be published.
Reviewer 2 Report
The authors have appropiately addressed all my concerns. I think the publication is suitable for publication.